# Debiasing Through Circuits: A Reproducibility Study in Mechanistic Interpretability

## Abstract

**Warning**: This paper includes discussion of stereotypes, biases, and toxic content for the purpose of improving AI safety; reader discretion is advised. Large language models (LLMs) achieve remarkable performance yet remain vulnerable to adversarial attacks. Mechanistic interpretability offers a promising avenue for diagnosing these weaknesses by identifying the circuits that drive model behavior. We reproduce and critically assess the pipeline introduced by García-Carrasco et al. (2024), which uses activation patching, gradient-based adversarial attacks, and logit attribution to locate vulnerabilities in a synthetic acronym prediction task for GPT-2 SMALL. While their approach provides an interesting toy example, we find incomplete circuit identification and limited adversarial effectiveness. To address these shortcomings, we apply edge attribution patching for more faithful circuit discovery, generalize their adversarial approach to multi-token inputs, and scale the analysis to a larger model, LLAMA-3.2-1B-INSTRUCT, on a more complex and socially relevant task: toxicity detection with a focus on name-related biases. We further introduce Differential Circuit Editing (DICE) to demonstrate how targeted interventions in the identified circuits can mitigate harmful behavior without compromising task accuracy, resulting in a bias reduction of 12.6% while improving accuracy of toxicity detection by 3.4%. [1]

## 1 Introduction

Although large language models (LLMs) have demonstrated state-of-the-art performance in a variety of natural language processing tasks, they remain susceptible to adversarial attacks (Zou et al., 2023; Abad-Rocamora et al., 2024). Understanding and mitigating these vulnerabilities is crucial for deploying LLMs in reliable real-world systems. One promising direction for vulnerability analysis is provided by the burgeoning field of Mechanistic Interpretability, which seeks to identify the internal circuits used by LLMs to perform tasks (Conmy et al., 2023; Lieberum et al., 2023b; Marks et al., 2024). But one area which has remained largely understudied is how adversarial attacks manifest as vulnerable components in the tasks' circuits.

With this in mind, García-Carrasco et al. (2024) proposed a methodology for identifying vulnerabilities related to a task and understanding the underlying mechanisms. Their pipeline involves generating a dataset, identifying relevant circuits through activation patching, performing gradient-based adversarial attacks on select parts of the input, and then using logit attribution to locate the circuit components affected by the vulnerabilities. They demonstrate their approach on an acronym prediction task for GPT-2 SMALL(Radford et al., 2019), illustrating how adversarially optimized samples reveal which letters are most likely to be misclassified, and how that misclassification can be traced back to specific components in the task's circuit.

While these findings are demonstrative, important limitations remain. The task tested by the authors is very simple, synthetic, and investigated on a small model, potentially restricting the generality of the approach. Despite promising progress, much of earlier circuit analysis research have been conducted on small models for relatively simple behaviors, such as induction heads (Elhage et al., 2021; Olsson et al., 2022), indirect object identification (Wang et al., 2022a), addition (Nanda et al., 2023; Quirke & Barez, 2023) and greater-than

---

[1]Source code and instructions for replicating our results are online at: `https://anonymous.4open.science/r/re-vulnerabilities-4107/`

(Hanna et al., 2023). More recent studies have shown progress in scaling circuit analysis to larger circuits and models, such as multiple choice question answering (Lieberum et al., 2023a). Although these works have undeniably improved our understanding of transformer circuits in tightly controlled scenarios, for the critical application of mechanistic interpretibility in AI safety (Bereska, 2024), model scale and task complexity are central to advancing the field. It is thus crucial to evaluate the authors' methodology on larger models for tasks of greater complexity and societal impact.

To bridge this gap, our work reproduces the methodology of García-Carrasco et al. (2024) while extending it to a more challenging setting: toxicity detection on LLAMA-3.2-1B-INSTRUCT, with a special emphasis on uncovering name-related biases in toxicity detection. Toxicity detection is inherently a challenging task with significant practical importance (van Aken et al., 2018; Pavlopoulos et al., 2020; Sheth et al., 2021; Cao et al., 2023). It is well known that certain user demographics and minority groups are disproportionately affected by biases in this task, and LLM systems deployed to moderate online content and mitigate harmful language must be robust and free of unintended biases (Vidgen et al., 2020; Dinan et al., 2020; Welbl et al., 2021; Cheng et al., 2023). Moreover, we specifically explore name-related bias in toxicity detection, recognizing that even subtle biases against certain demographic groups can cause substantial harm if left unchecked (Borgesius, 2020; Bender et al., 2021). By scaling up the model size and focusing on a task that is both more difficult and highly impactful in real-world applications, we provide a more comprehensive testbed for evaluating the generality and robustness of the authors' methods. In addition, this broader setting allows us to investigate how model biases might arise from or be amplified by specific circuit components, thereby offering insights into how targeted interventions could mitigate these harmful behaviors.

Our main contributions towards the reproduction and extension of the work can be summarized as follows.

**Reproducibility evaluation.** We replicate the authors' circuit identification method, adversarial sample generation algorithm, and vulnerability-locating strategy. This confirms the reliability of their approach on the original acronym prediction task.

**Faithful circuit identification.** We employ edge attribution patching (EAP) methods to determine faithful circuits that cannot be obtained solely through activation patching of attention heads. This provides deeper insights into the model's internal mechanisms.

**Generalization of the adversarial sample generation algorithm.** We extend the adversarial sample algorithm to work with multitoken samples, moving beyond single-token cases and thus making it more broadly applicable to other tasks.

**Scaling to a complex task and larger model.** We apply investigated methodologies to the more complex toxicity detection task on the LLAMA-3.2-1B-INSTRUCT model (Meta, 2024). This investigates the applicability of these methods in a realistic, impactful setting.

**Introducing a method for debiasing via circuits.** We introduce Differential Circuit Editing (DICE), a systematic method for correcting task targeted vulnerabilities in faithful circuits. The method highlights that scaling values of specific edges is an effective technique for mitigating the vulnerabilities associated with a specific circuit while preserving performance on the relevant task.

The importance of this reproduction can be framed in the discussion of AI safety. Safe AI must be robust to adversarial attacks because the ability to defend against small, crafted perturbations is foundational for solving more complex safety issues. There is a symbiotic relationship between interpretability and adversarial robustness. Through analyzing adversarial vulnerabilities, we can make models both more interpretable and robust (Bereska, 2024). Adversarial examples are shown to be interpretability tools (Casper et al., 2022; Dong et al., 2017). Indeed, Tomsett et al. (2018) highlight that the existence of adversarial examples demonstrate that class boundaries do not align with the intentions of model builders, leading to bias. Hence, we can better understand a model's flaws using such examples.

Although some approaches have addressed model vulnerabilities (García-Carrasco et al., 2024; Sadria et al., 2023), in general there is a paucity of work in this area. The original study, while successfully introducing an approach for understanding adversarial vulnerabilities, has some shortcomings. The setting used to demonstrate the methodology is impractical; the authors employ a small model for an overly simplistic task.

The original work only performs activation patching on heads rather than identifying the full circuit. Our contention is that the introduced framework calls for a study that is more in line with what is used in practice. Furthermore, the authors do not provide a method for mitigating the vulnerabilities that they identify, as required to improve adversarial robustness. So, by reproducing the original study, and extending it to a more complex and societally relevant task, we make a contribution towards improving adversarial robustness and AI safety in general. We highlight both the broader applicability of vulnerability detection in circuit analysis, and the utility these methodologies provide for bias mitigation.

## 2 Scope of reproducibility

García-Carrasco *et al.* propose as a main contribution their methodology for locating vulnerable circuits. The claims made by the authors can be summarized as follows.

**Claim 1: The circuit associated with a task can be identified by the authors' chosen MI techniques.** By conducting activation patching, the authors first identify the most important components for the acronym prediction task.

**Claim 2: Adversarial samples can be generated for a task using the authors' proposed algorithm.** The authors propose a gradient-based adversarial attack, Algorithm 1 can be found in Appendix D.

**Claim 3: Using the adversarial samples, the vulnerable components of a circuit can be located using MI techniques.** Samples generated from the adversarial attack are used to isolate vulnerable heads from within the identified circuit for the given task.

Our contention is that the original methodology should be robust and extensible to other models and tasks to truly benefit the field. Modern machine learning tasks require models with hundreds of billions of parameters Chowdhery et al. (2022). We question how the methodology will scale, in particular, how the vulnerable components of a circuit will manifest themselves with a larger number of parameters. So, a secondary claim is that the three primary claims must be able to be applied for any LLM and for any task. For this reason, we scale to LLaMA-3.2-1B-Instruct from GPT-2 Small. In fact, it was necessary to utilize a larger model for the toxicity prediction task as GPT-2 Small could not achieve sufficient performance.

## 3 Methodology

### 3.1 Reproduction

Using the authors' released code [2], we reimplement their approach of circuit identification via activation patching to identify circuits. Activation patching involves replacing the activations of a given component with the activations obtained by running the model on a corrupted prompt. We note here that activation patching with corrupted examples (resampling ablation) is not the only practice, with recent study indicating that the preferred type of ablation should be task dependent (Miller et al., 2024). For our reproduction, we follow the original authors in implementing resampling ablation. For the acronym prediction task, the corrupted prompt is equivalent to the original prompt, with the exception that the third word is replaced by another word beginning with a different letter. With the corrupted activations, we patch all layers and heads. A large reduction in logit difference, as shown in Equation 1, indicates that the component is necessary for the task and therefore present in the circuit.

García-Carrasco *et al.* introduced Algorithm 1 for adversarial sample generation in the discrete data domain based on Wen et al. (2023)'s PEZ algorithm for prompt tuning. Starting with a correctly classified sample, the algorithm computes the difference between the logits of the labels as an objective. Its gradient is backpropagated through the model up to the embeddings of all tokens in the input prompt. In the original approach, the authors calculated the difference between the logit of the correct letter for the acronym and the logit of the incorrect letter with the highest score. Importantly, the model parameters are not updated during this process. Instead, the input embeddings are treated as changeable, allowing for the computation

---

[2]https://github.com/jgcarrasco/detecting-vulnerabilities-mech-interp

of gradient-based adjustments. These gradient-based adjustments are calculated via backpropagation from the adversarial loss, designed to maximize the difference between the logit for the correct class and the highest logit for any incorrect class. The assertion is that, through these small gradient-based changes, a misclassified sample can eventually be generated. This misclassified sample is then be mapped back to meaningful tokens in the vocabulary.

These samples are then used in a series of logit attribution experiments. Logit attribution is described in Equation 2. In these experiments, for each head in the model, we cache the output and project the vectors into the direction of the logit difference. So, if a head outputs a negative value for a sample, it indicates that the highest logit for any incorrect class outweighs the logit of the correct class. This implies the head misclassifies that sample, allowing us to determine that it is vulnerable.

$$\text{logit diff}_i = \text{logits}_{a_i} - \max_{a_j \in L \setminus \{a_i\}} \text{logits}_{a_j} \tag{1}$$

where $L$ is the set of possible answers, in this case the letters of the alphabet, $a_i$ is the correct answer and $\text{logits}_{a_j}$ is the logit associated to the token $a_j$.

$$\text{logit attr}_k(h_{ij}) = W_U[k] \cdot h_{ij} \tag{2}$$

where $W_U[k]$ is the kth column of the unembedding matrix, with $k$ being the token number. $h_{ij}$ is the output of the ith attention head of the jth layer.

## 3.2 Datasets

For replicating the original task, we reuse the dataset of three letter acronyms made available by the authors. Each prompt formed using the dataset has three words, followed by two capital letters (e.g. "The Chief Executive Officer (CE"). The third letter in the dataset is used as the answer to the prompt. Each of the words in the dataset were selected as single tokens by GPT-2 SMALL to facilitate the subsequent adversarial attack.

Additionally, we create a custom dataset for the toxicity prediction task. The dataset combines a popular names by country dataset scraped from Wikipedia[3] with the PolygloToxicityPrompts dataset from which we obtain sentences with varying levels of toxicity (Jain et al., 2024).

For toxicity scores in the range of 0 to 1, we split the entries of the data into 4 buckets of 0.25, and randomly sample 48 entries to balance toxicity scores when generating our dataset. For each toxic sentence, we substitute 36 different names from the names dataset, originating from 12 different subregions, 3 names per subregion, with subregions being Western Europe, Eastern Europe, North Africa, etc. The final dataset is composed of 5 data splits, each of 1728 entries, generated with 5 different random seeds. Although the dataset is split into five subsets to enable robust evaluation across random seeds, some experiments employ only a subset of these splits due to computational constraints. The general structure of each of our data is: "[NAME] wrote: 'toxicity_sentence' ", where toxicity_sentence is an entry obtained from PolygloToxicityPrompts. Each prompt features a two-shot preamble before the data point. The binary target label of each datapoint is derived from the toxicity score of the sentence, where sentences with toxicity scores lower than 0.5 are labeled as non-toxic, and those with scores over 0.5 are labeled as toxic. The way we have designed the toxicity prediction task naturally introduces our name-related bias task, since the same toxicity_sentence is evaluated across multiple names. This setup introduces the potential for bias, where different names paired with identical sentences may be predicted as more or less toxic.

## 3.3 Faithful Circuit Identification

To motivate the introduction of faithful circuit identification methods, we follow Hanna et al. (2024) to formalize related concepts. We provide a more detailed introduction to the used methods in the Appendix Section A. An identified circuit, composed of a computational subgraph of the model, is said to be faithful

---

[3]https://github.com/sigpwned/popular-names-by-country-dataset?tab=readme-ov-file

when all edges outside the circuit can be corrupted while retaining the original task performance of the full model. Formally, corruptions are performed with ablations that patches the activations of a clean run of the model using clean inputs, with a corrupted run of the model using corrupted inputs. This technique of activation patching introduced by Meng et al. (2022) was applied to identify neuron activations responsible for factual predictions. In sum, if the change in activation on all component (head, neuron, layer, etc) outside the identified circuit does not impact the performance of the model, the circuit is said to be faithful for the task. To measure this performance, typically the metric of logit difference between the logits of two possible next tokens is used (logits of True/False for our task of toxicity detection, this is our metric $m$).

In an identified circuit, the model components outside of the subgraph are ablated, but to identify which components to ablate for a faithful circuit, typically a circuit identification method is used, instead of ablating the components individually since individual intervention scales poorly with model size. For circuit discovery, we employ edge attribution patching (EAP) introduced by Nanda (2024a); Syed et al. (2023), and edge attribution patching with integrated gradients (EAP-IG, along with its KL divergence alternative EAP-IG-KL) introduced by Hanna et al. (2024). The score of an edge in the computational subgraph is calculated to be proportional to their influence on the gradient. Further details on these methods are presented in Appendix A. Our faithfulness metric is then normalized against the same metrics evaluated on the full model with clean and corrupted inputs using equation Equation 3.

$$\text{normalized\_faithfulness} = (m - b')/(b - b') \tag{3}$$

where $m$ is the circuit's performance, $b'$ is the whole model's performance on corrupted input, and $b$ is the whole model's performance on clean input. We apply these methods for circuits discovery of the toxicity prediction task and a name-related bias task based on the dataset introduced in Section 3.2. In the toxicity task, the difference between clean and corrupted inputs are the toxicity sentences, while the names remains unchanged. For the name-related bias task, the toxicity sentences are unchanged, but names are changed between clean and corrupted inputs. There is not an existing standard on the amount of performance that should be recovered by a circuit for it to be considered faithful. Hanna et al. (2024) aimed to find minimal circuits (1-2% of edges) and recover at least 85% of the model performance, Yao et al. (2024) found knowledge circuits in GPT2-Medium that recovers 70% of the original performance with a subgraph of less than 10%. For our task we aimed to recover as much performance as possible with at most 10% of the edges.

### 3.4 Adversarial Sample Generation

We extend Algorithm 1 for multitoken adversarial sample generation (shown in Appendix D), reusing the same loss function. This was necessary because the original algorithm is only capable of finding single-token adversarial samples, and there are very few single-token names in our application. For a sample of size $n$, we map an adversarial sample of size $n$, restricting our search space such that these samples must match exactly in token length. For example, an adversarial sample of size 2 cannot be comprised of two independent samples of size 1. When projecting the sample to real embeddings $p$, we match the virtual embeddings $p'$ with each possible set of real embeddings using average dot product similarity as in Equation 4. The selected real embeddings are the ones that maximize this metric.

$$p = \arg\max_p \left( \frac{1}{l} \sum_{i=1}^{l} p_i \cdot p'_i \right) \tag{4}$$

### 3.5 Locating and Understanding Vulnerabilities

The authors' approach to identifying vulnerable attention heads involved analyzing their independent contributions using adversarial samples. Each attention head's activation was separately projected into the residual stream, treating it as if it were the model's output. This modified residual stream was then passed through the final layer normalization and projected into the logit space. This allowed the authors to compute the difference between logits for the target tokens of interest. Their rationale was that this method would reveal the individual contribution of specific attention heads to the model's predictions, thereby identifying those most responsible for adversarial vulnerability.

### 3.6 Bias Correction

To mitigate name related bias as described in Section 3.2, we propose Differential Circuit Editing (DICE), a method for correcting task-specific vulnerabilities in faithful circuits. It operates by analyzing two circuits responsible for distinct model behaviors and identifying edges that are present in one circuit but absent in the other. Once these edges are identified, various interventions can be applied to mitigate the unwanted behavior of the model.

In our task, we use DICE as a bias mitigation approach by leveraging two identified circuits: the toxicity circuit and the name-bias circuit described in Section 3.3. Specifically, we identify the edges that are present in the bias circuit but absent from the toxicity circuit and selectively scale down their magnitude. We hypothesize that this approach will reduce the influence of attention heads responsible for name-related bias while preserving the model's performance on the toxicity prediction task. Figure 1 illustrates an example of this bias correction mechanism.

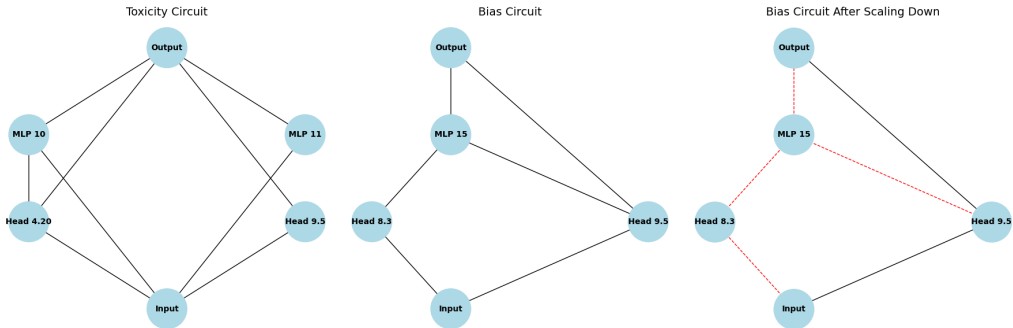

Figure 1: Example of the bias correction mechanism with the scaling down the magnitude of the edges present in the bias circuit but not in the toxicity circuit (red, dashed edges on the third subgraph).

To assess bias reduction, we introduce a metric based on the z-normalized logit difference between "true" and "false" tokens. Z-normalization transforms values into a standard normal distribution by subtracting the mean and dividing by the standard deviation, ensuring comparability across different scales. We group the this logit differences by 'toxicity_sentence' (i.e., across different prompt templates) and compute the standard deviation within each group (Equation 5). The final bias score is then obtained by averaging these standard deviations across all prompt templates. This metric provides an interpretable measure of how consistent the model's predictions are across different names in prompts. A higher bias score indicates greater sensitivity to name changes, suggesting potential bias in the model's decision-making. A score of zero would imply that altering names in the prompt does not affect the predicted logits, meaning the model treats all names neutrally.

$$\text{bias\_score} = \frac{1}{N} \sum_{i=1}^{N} \sigma_i \tag{5}$$

where $N$ is the number of 'toxicity_sentence' groups and $\sigma_i$ is the standard deviation of the z-normalized logit differences within the ith toxicity sentence.

To further analyze bias reduction, we introduce a second approach that examines the variation in normalized logit differences across different regions. We first compute per prompt the variation score by subtracting the mean normalized logit difference within each prompt template (Equation 6). By grouping these variation scores per region (Equation 7), we obtain the mean deviation of logits for each region. Instead of only measuring prediction inconsistency, this method helps us determine if the model systematically favors certain groups over others. If all regional scores are close to zero, it indicates that differences in predictions across names are purely random noise, meaning the model is inconsistent but not favoring any specific group. On the other hand if the regional scores exhibit a clear trend (e.g., some regions have consistently higher or lower

scores), this suggests that certain names are systematically influencing the model's predictions, revealing a potential bias.

$$\text{variation\_score}_i = \text{logit\_diff}_i - \sum_{j \in J} \frac{\text{logit\_diff}_j}{|J|} \tag{6}$$

where $\text{logit\_diff}_i$ is the z-normalized logit difference for sample $i$ and $J$ contains all indices $j$ of samples with the same toxicity_sentence as prompt $i$.

$$\text{region\_variation\_score}_r = \sum_{i \in R} \frac{\text{variation\_score}_i}{|R|} \tag{7}$$

where $\text{region\_variation\_score}_r$ is a score for region $r$ and $R$ contains all indices $i$ of samples with the name from region $r$.

## 4 Experiments

Our work involves five experiments. The first is a reproducibility experiment, in which we attempt to reproduce the original author's methodology to the best of our ability. The second experiment compares activation patching to faithful circuit identification. The third determines the generalizability of the adversarial sample generation algorithm. In the fourth experiment, we contrast vulnerability detection with the authors' projection-based method with the faithful circuit method on the bias detection task. Finally, in the fifth experiment we attempt to correct the bias while maintaining performance in the toxicity prediction task.

**Experiment 1** reproduce the original work of the authors using their codebase. In accordance with the original paper, we use GPT-2 SMALL for the task of predicting the third letter of the acronym corresponding to the third word in the prompt. We replicate the circuit identification procedure to locate the relevant attention heads, determine the logit difference at each head, and conduct their proposed adversarial attack on all attention heads in the model to locate the vulnerable components.

**Experiment 2** test the faithfulness uncovering the true circuit by employing the LLAMA-3.2-1B-INSTRUCT model in the task of toxicity prediction in the samples from the dataset described in Section 3.2. We use their chosen method of activation patching, and compare the logit difference of the clean and corrupted samples on the tokens of the labels ("true" and "false"). Then we employ EAP methods described in Section 3.3 as an alternative method to identify the relevant faithful circuit and subsequently test the faithfulness of this circuit by performing resampling ablations on all components which are not present in our circuit to determine whether the performance on the toxicity prediction task changes. Finally, we contrast our circuit with the components found using the original method.

**Experiment 3** aim to evaluate the generalizability of the authors' proposed method for finding adversarial samples. The approach described in Section 3.1 relies on leveraging gradients. This experiment seeks to determine whether it is effective for longer prompts and more complex tasks. Intuitively, as the length of the sentence increases, the gradient is distributed across a larger number of tokens in the prompt. Although the issue of gradient magnitude can be addressed by increasing the learning rate, challenges arise in scenarios where we aim to generate meaningful samples by allowing changes to only a subset of tokens using masking. In such cases, much of the gradient could be allocated to masked tokens, leaving insufficient gradient flow for unmasked tokens to induce a label change. To test this hypothesis, we analyze the gradients generated during the authors' adversarial sample generation process on the acronym task using the GPT-2 SMALL model and compare them with the gradients obtained from name-related bias task using the LLAMA-3.2-1B-INSTRUCT model.

**Experiment 4** evaluate the proposed method for identifying vulnerable attention heads. The approach assumes that a head's activation, when projected into the residual stream, represents its entire contribution to the model's output. However, this simplification neglects interactions between heads across layers, raising questions about the completeness of the method. A key concern is whether this technique captures all

vulnerable heads or if some remain undetected due to redundancy in the model. Prior research suggests that ablating certain components can lead to structural changes in the model's behavior, meaning that a supposedly vulnerable head may be compensated by another Wang et al. (2022b). To assess the validity of authors' method of activation patching, we compare it to the edge attribution patching methods described in Section 3.3, using adversarial samples from Experiment 3 for the name-related bias task. If both methods identify the same heads as influential, it would strengthen the confidence in the projection-based technique.

**Experiment 5** extend the methodology from vulnerability detection to debiasing. We utilize the toxicity circuit from Experiment 3 and the bias-vulnerable circuit from Experiment 4, and apply DICE as outlined in Section 3.6. We evaluate this approach on five sampled datasets by measuring the accuracy of toxic sentence classification, introduced bias score (Equation 5) and comparison across variation scores per region (Equation 7).

## 5 Results

### 5.1 Results reproducing original paper

**Claim 1** The authors claim that activation patching is sufficient to identify and understand the underlying circuit responsible for a given task. We reproduced the activation patching plot in Figure 8a. The results closely align with those reported by the authors, with the exception that, in our case, head `9.9` exhibits a lower logit difference value compared to the original paper. Even though this analysis provides us with the important heads, to fully validate the author's claim, we perform activation patching (Section 5.2) and circuit identification method (Section 5.2) on our task and bigger model.

**Claim 2** One of the main contributions by the authors is their application of the adversarial algorithm for determining the vulnerable components of a circuit. We highlight Figure 2 as it differs from the original paper's findings. The chart is dependent on the randomly generated adversarial samples, so divergence is expected. Rather than using a subset of the original distribution as the original authors' may have done, we elect to use the entire original distribution when calculating each $p_{org}^i$ to maximize consistency. For reference, $\Delta p = (p_{adv}^i - p_{org}^i)/p_{org}^i$ where $p_{adv}^i$ is the probability of samples with the third word beginning with letter $i$ in the adversarial distribution, while $p_{org}^i$ is the same, but for the original distribution from the dataset.

Through the reproduction of their work, we determine that their algorithm works well on the original task, with head `10.10` consistently misclassifying the letter `A` as the letter `Q`. The letter A is the most often misclassified. Indeed, we find that samples with a third word beginning with A are around 13 times more frequent in the adversarial sample set. Overall, we find this claim to be partially verified. We elaborate on this conclusion in Section 5.2.

**Claim 3** We generated Figure 8b to examine logit attribution for each attention head on adversarial samples containing the letter A. Three key heads were identified: `10.10`, `9.9`, and `8.11`. These are the same heads found in Figure 8a, indicating that these components are indeed vulnerable. In this respect, our results are consistent with the claim made by the authors, but we subsequently determine that this claim is problematic.

### 5.2 Results beyond original paper

**Activation Patching** Figure 3 (left subplot) shows the activation patching plot for the toxicity prediction task using LLaMA-3.2-1B-Instruct on one split of our dataset. This identified several particularly important attention heads: `4.2`, `8.2`, `6.6`, `9.20`, `7.21`, and `8.24`. To consider a head important, we look at the absolute value of logit difference for the head, regardless of the sign of the value. Notably, the toxicity prediction task appears to involve a greater number of significant heads compared to the acronym prediction task. This may be attributed to two factors. First, unlike acronym prediction, which relies on a single token, toxicity prediction requires a more complex and abstract understanding of the entire sentence. Second, the increased model size and complexity may lead to a more distributed representation, where individual atten-

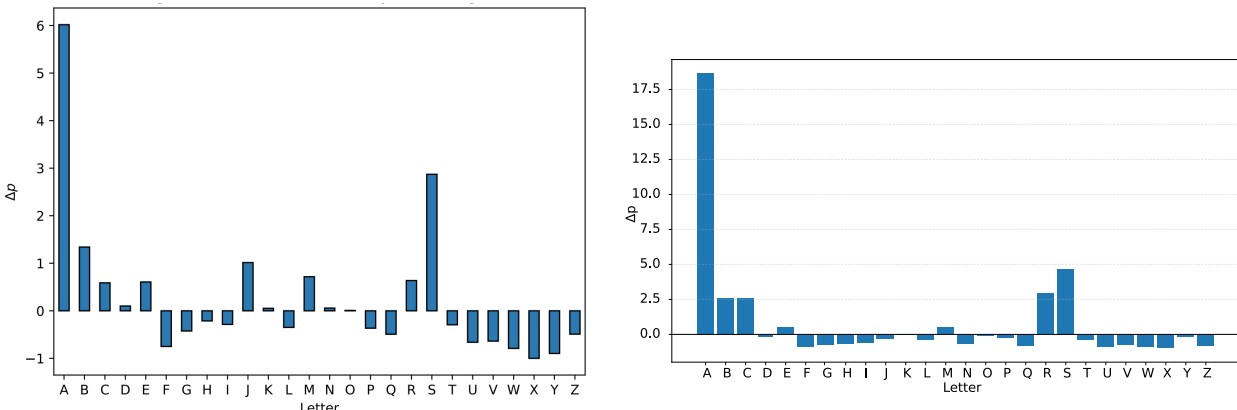

Figure 2: A comparison between the figure obtained in the original paper (left) and the distribution we obtained (right). The plots specify the distribution of the words of the dataset that begin with each letter vs. the distribution of generated adversarial acronym in terms of the initial letter of the third word.

tion heads contribute less to overall model performance, as perturbations in one head can be compensated by others.

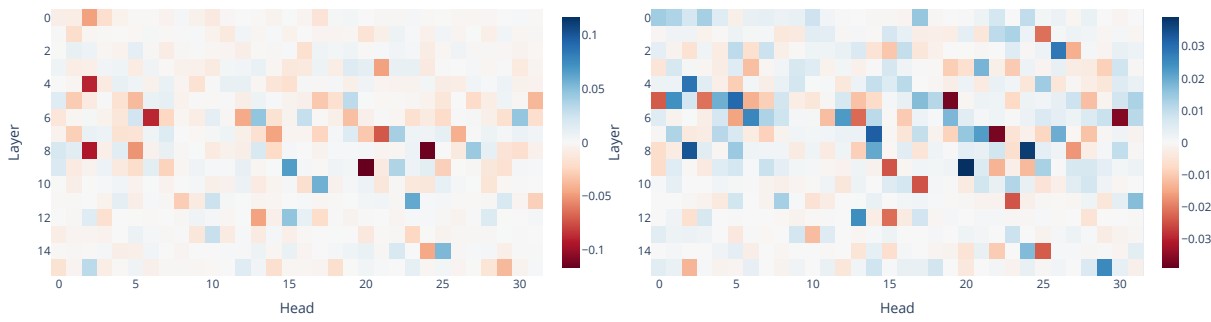

Figure 3: **Left**: Activation patching plot for toxicity task for all heads and layers in LLaMA-3.2-1B-Instruct. **Right**: Sum of normalized edge scores for all heads and layers in a 91% faithful circuit identified with EAP-IG for the toxicity task.

**Circuit Identification using EAP** The right subplot in Figure 3 displays the summed normalized edge scores for all attention heads on a circuit for the toxicity task identified with EAP-IG. The experiment was performed on one split of the dataset. The specific circuit has 18922 edges (9.6% of edges) and is 91% faithful [4]. This visualization can be interpreted as a representation of head importance. Similar to activation patching, we consider a head important when the absolute value of the score for the head is high. When comparing the two methods, we see that while activation patching identifies identical heads as the faithful circuit method (`4.2`, `8.2`, `6.6`, `9.20`, and `8.24`), a significant number of important heads are not identified via activation patching (`5.5`, `5.19`, `7.14`, `7.22`, and `6.30` to name a few). While the activation patching method provides an overview of the most important model components related to the task, it fails to capture the full mechanism and does not offer insight into the interactions between model components.

**Gradient Flow** From measuring within the iterations of Algorithm 1, we obtained gradient values at the embedding level, normalized for each task. To better understand the distribution of gradient flow, Figure 4

---

[4]The graphs for evaluating normalized circuit faithfulness of discovered circuits can be found in Appendix E

illustrates which tokens received the most gradient flow in an example sentence. The gradient score, which represents the flow, is calculated as the absolute mean of the gradient on the embedding of a specific token. By averaging these gradient scores across iterations, we can observe where the majority of the flow accumulates for the given sentence. In the example using the authors' code, we observe a high gradient for the third word ("Jeep"), which aligns with expectations. However, in our case, the gradient flow is concentrated on tokens in the two-shot preamble, with only a small portion directed toward the name token ("Morales"). Additional plots of example sentences are provided in Appendix C, offering more detailed visualizations. This trend is consistent across all samples. Figure 5 shows the mean gradient scores, evaluated on the 3 data splits, for the masked tokens against the unmasked tokens, i.e., the tokens we aim to modify, across iterations. In the authors' task, the unmasked tokens, such as the third word in the acronym prediction, consistently receive higher gradient scores.

In contrast, in our task, the masked tokens generally receive higher gradients than the name tokens. This indicates that swapping names is not an effective choice for adversarial sample generation. The algorithm would rather modify some of the masked tokens, i.e. the other tokens in the prompt, than the names to change the classification. However, altering these tokens can unintentionally change the sentence's actual toxicity. For example, if a profane word is replaced, the sentence might no longer be classified as toxic, but its ground truth label should also be updated to not toxic, making it an invalid adversarial sample. In the original task, it is made simple which token the adversarial sample generation algorithm should modify. This aligns with our experimental results: from only 384 data samples in the original task, we generated 1,364 adversarial samples — a rate of 355%. In contrast, in our task, we produced just 311 adversarial samples from 1,371 inputs, indicating a markedly lower generation rate of 22%, almost the reverse trend observed in the original task. However, in general this is an open problem. So while the adversarial sample generation works well for the original task, our findings indicate that the method lacks robustness to generalize.

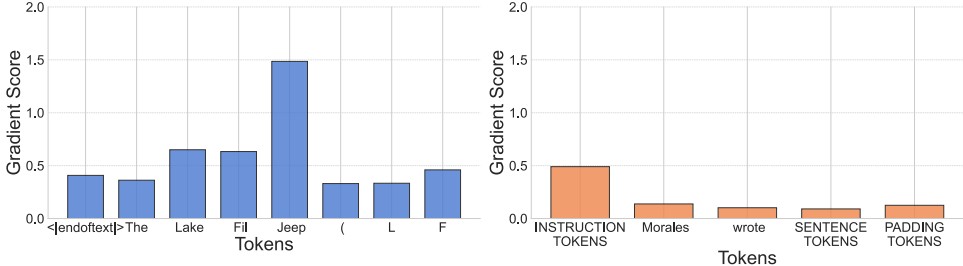

Figure 4: Example of normalized absolute mean gradient for tokens in a sentence. **Left**: for acronym prediction task. **Right**: for bias detection task, *instruction tokens*, *sentence tokens* and *padding tokens*, representing their average gradient scores. Sentence tokens are the tokens of the toxic sentence.

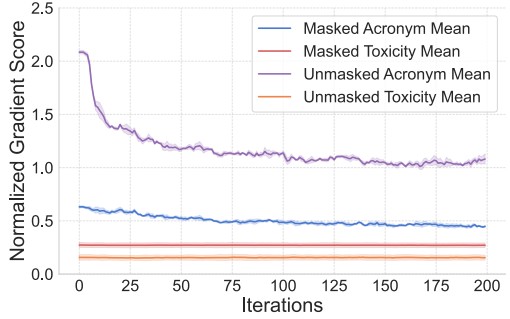

Figure 5: Presented absolute mean gradient value over iterations for masked and unmasked tokens generated using adversarial generation method. For acronym prediction and toxicity prediction tasks, gradients have been normalized individually for each task to ensure fair comparison.

**Finding Vulnerable Components** Our result for the bias detection task shown in Figure 6 exhibit significant differences between the authors' method and the bias circuit. This circuit is significantly smaller (6908 edges, 3.5% of edges) compared to the toxicity circuit as only the name tokens are changed between clean and corrupted inputs, and is only 70% faithful. The vulnerability map, presented on the right subplot, is computed by summing the contributions of all edges for each node of the bias circuit. The authors' approach highlights a few heads with high-magnitude scores, whereas the circuit identification method reveals sparse clusters of vulnerable heads, all with significant absolute values, distributed across layers.

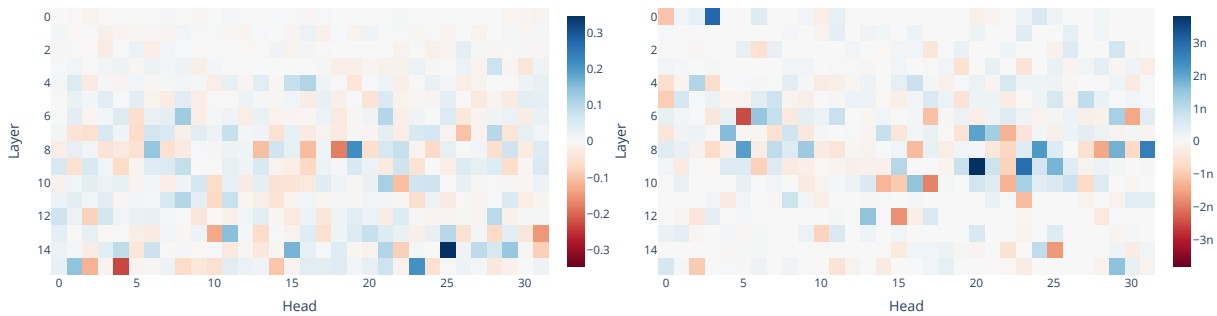

Figure 6: **Left**: Vulnerable components identified by the authors' method for the bias detection task, with scores based on the final logit difference between the correct and incorrect tokens. **Right**: Sum of normalized edge scores for all heads and layers in circuit identified with EAP-IG-KL for the name bias task.

**Bias Correction** The impact of scaling on model performance shown on Figure 7 demonstrates that for scaling factors between 0.9 and 0.7, we successfully reduced the bias metric (equation 5) without compromising overall model accuracy. In fact, we were able to slightly improve accuracy through this approach. By analyzing the region variation score (equation 7) in Table 1, we observe a clear trend of bias favoring specific regions. Names from Southern Europe tend to shift predictions toward the toxic label (token "true"), whereas names from South Asia decrease toxicity predictions, pushing the model toward the non-toxic label (token "false"). Applying a scaling factor of 0.7 effectively reduced these biases, moving regional influence closer to neutrality. Notably, reductions observed for Southern Europe (-18.8%) and South Asia (-30.6%), i.e. the regions with the biggest bias, indicate a significant correction in the model's regional sensitivity.

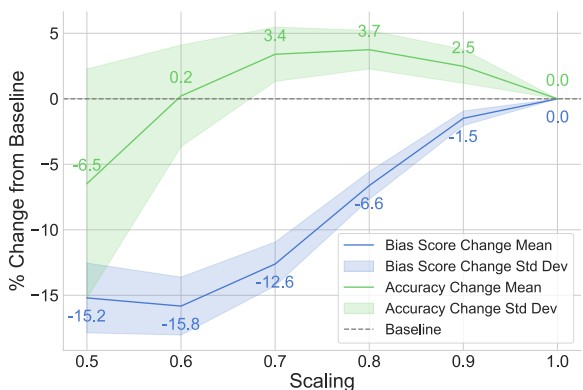

Figure 7: Results of debiasing for different scaling factors applied to the selected edges as a percentage change of metrics from baseline (default model run). Accuracy metric is overall accuracy on the dataset, score metric is our metric of standard deviation grouped by prompt template. The results were calculated across 5 generated datasets.

| Region | Std. Dev. Score Baseline | Std. Dev. Score Scaling | Percentage Change |
|---|---|---|---|
| Southern Europe | 0.114 (±0.022) | 0.092 (±0.016) | -18.8 % (±2.2 %) |
| South Asia | -0.232 (±0.02) | -0.161 (±0.016) | -30.6 % (±3.2 %) |

Table 1: Mean performance with and without scaling to 0.7 selected edges of the model with standard deviation in the brackets. Percentage change was calculated between scaled and baseline scores on 5 sampled datasets and averaged. The full table is made available in Appendix F.

## 6 Discussion

The authors' method for identifying circuits can be used to determine the attention heads responsible for toxicity prediction but does not provide a complete identification of the entire circuit. Heads are isolated components of the model, so while they provide a local view of model behavior, they do not capture the holistic interaction between components within the model. They may highlight areas of fragility, but they fail to reveal how those heads function together to process information. Using different circuit identification methods produces different results, even on the same dataset. This highlights the importance of ensuring the accuracy and reliability of the identified circuit. This leads us to the conclusion that **Claim 1** is **invalid**.

Finding adversarial samples for our task of bias identification is particularly challenging. Identifying a toxic prompt that is classified correctly for one name but misclassified for another heavily relies on the inherent biases within the model. Consequently, the total number of generated samples across tasks is not directly comparable. However, the tendency of gradients to flow to masked tokens appears to be an inherent limitation of this method, especially for more complex tasks. This directly impacts the method's efficiency and effectiveness.

As well, the projection step introduces further challenges. Even if embeddings are successfully altered to cause misclassification, projecting them back into the vocabulary can lead to embeddings that are still classified correctly. So, the choice of vocabulary and its size influence the method's success. For example, given a name-related bias task with a vocabulary containing only one name, any embedding changes would still be projected to the same name, preventing the generation of valid adversarial samples. While the iterative process ensures that adversarial samples remain close to the original inputs, the method's drawbacks, inefficiency and difficulty in producing samples, were particularly evident in our task. We state that authors' **Claim 2**, is **partially valid**. The method has limitations and works only under specific conditions.

Our results suggest that the authors' method for identifying vulnerable heads in transformer models may be incomplete. Their approach, which projects each head's output into the residual stream and treats it as its sole contribution, highlights only a few isolated heads with high-magnitude scores. In contrast, our circuit-based method, which sums edge scores from the EAP methods by Hanna et al. (2024) to capture interactions between components, reveals a broader pattern of vulnerability. Rather than a few distinct heads, we observed clusters of susceptible heads spanning multiple layers. Our findings for LLaMA 3.2 in Figure 6 suggest that vulnerability is distributed rather than localized, with some heads compensating for others when ablated. This raises concerns about whether the method captures all vulnerable components or if redundancy obscures some. Therefore, we conclude that the authors' **Claim 3** is **invalid**.

Our DICE method successfully demonstrates that modifying specific circuit edges, in this case through scaling, can selectively suppress undesired biases while maintaining task-relevant behavior. We identified regional name biases (Table 1) and effectively mitigated their influence. The small gains in accuracy are expected as decreasing the bias reduces the undesirable contributions to the logits of tokens "true" and "false". If these are significant enough, they can negatively affect the classification. This highlights the potential of targeted circuit interventions as a technique for improving fairness and accuracy. Future research should explore the generalizability of this method across different tasks and model architectures.

### 6.1 Reproducibility

Reproducing the authors results were made easy by the availability of their code. Claims 1, 2 and 3 were the easiest to validate. Their code could be executed with minimal modifications. We were unable to reproduce the original task on a different model (LLaMA-3.2-1B-Instruct) due to the acronym tokenization being irregular with the model. In the case of GPT-2 Small, the words involved in the acronyms were chosen to represent three distinct tokens, whereas for LLaMA-3.2-1B-Instruct, each of these words may have a larger number of tokens, making it challenging to create such a dataset. Additionally, the LLaMA model tokenizes the acronyms differently, with the first two letters of the acronym often being represented by one token rather than two. These requirements compelled us to reproduce the methodology on a new task for testing its generalizability when scale to larger models. Adapting the code to fit the VRAM constraints of the A100 GPU for the new model and task was challenging due to the increased number of model parameters and longer input prompt lengths. To address this, we implemented a mechanism for consistently saving and reading files from memory rather than storing them in RAM. In addition, we reduced the batch size from 50 to 16. The original authors' code does not appear to be scalable for models and tasks with higher GPU memory requirements.

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

## A   Mechanistic Interpretibility

In this section we define the terminologies discussed in this work precisely and provide a more detailed introduction to the mechanistic interpretibility techniques used. We generally follow the definitions described by Nanda (2024a) and Hanna et al. (2024). In our study, we refer to circuits for a given task as being the minimal computation subgraph of the model whose behavior is faithful to the whole model's behavior on the task. It is a digraph linking individual nodes or components, such as specific attention heads, with edges, i.e. where a node's output flows. A faithful circuit then is a circuit for which we can corrupt all model edges outside the circuit while retaining the model's original task performance.

Techniques for ablating model components include methods like zero ablation that patches the activation with zero (similar to dropout at training time), mean ablation that patches the activation with the average activation over some data distribution (typically training data), and resampling ablation that replaces activation with a randomly chosen input.

Both edge attribution patching (EAP) and edge attribution patching with integrated gradients score edges in the computation graph by approximating the change in loss caused by corrupting an edge. Given an edge $(u, v)$, EAP approximates this change in loss by calculating the dot product between the difference of corrupted and clean activations $(z'_u - z_u)$ (where $z'_u$ and $z_u$ are the corrupted and clean activations at node $u$ respectively), and the gradient of the loss with respect to the input of $v$. The scored edges are then pruned based on a greedy algorithm to obtain the final circuit.

EAP-IG measures edge importance by combining EAP with Integrated Gradients, which addresses the zero gradient problem of EAP. When a model's internal activation has zero gradient at the input point, that activation will not contribute to the attribution in EAP, even if the activation has a non-zero gradient at the corrupted input point and the difference in activations is significant. EAP-IG resolves this by accumulating gradients along the straight-line path from the corrupted input to the clean input. Mathematically, the EAP-IG score is defined as:

$$(z'_u - z_u)\frac{1}{j}\sum_{k=1}^{j}\frac{\partial L(z' + \frac{k}{j}(z - z'))}{\partial z_v}$$

where $j$ is the number of integration steps, and $L$ is the loss function. The integral is approximated as a sum over $j$ steps along the interpolation path between corrupted and clean points. The method uses blended inputs during the integration process, ensuring non-zero gradients unlike standard EAP which may encounter zero gradients when evaluating only at the clean input point. This approach provides more robust edge importance scores, especially in cases where models exhibit significantly different behaviors between corrupted and clean inputs. EAP-IG also allows for the use of various loss functions, including Kullback-Leibler (KL) divergence (EAP-IG-KL).

For a detailed introduction to mechanistic interpretibility concepts, Nanda (2024b) provides a comprehensive glossary.

## B   Reproduced Figures

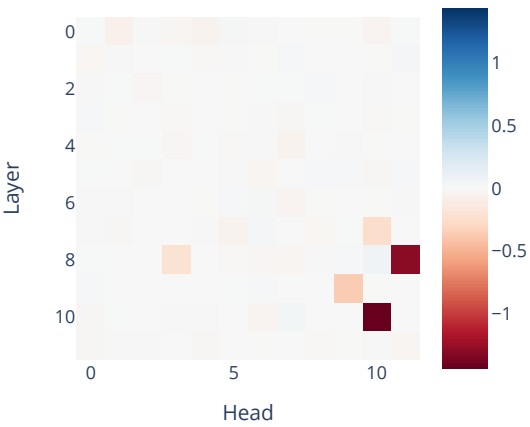

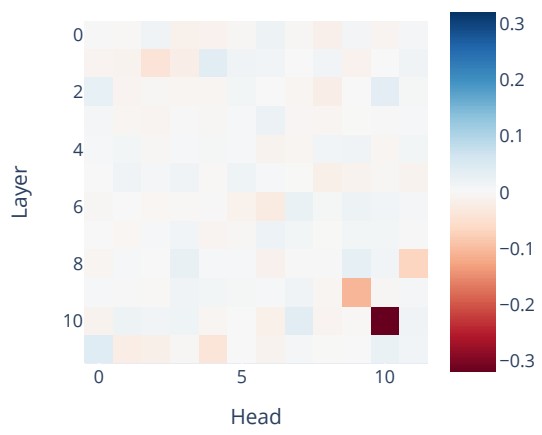

(a) Variation in logit difference when patching different heads on GPT-2 Small.

(b) Logit attribution for every attention head on adversarial samples with the letter A. This attribution is obtained by projecting into the logit difference direction.

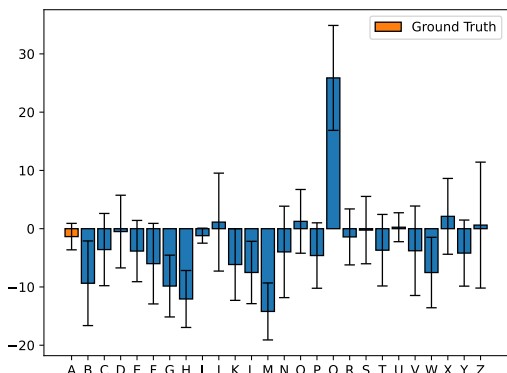

(c) Logit attribution of head `10.10` on adversarial samples with the letter A. This attribution is obtained by projecting into the directions of the different capital letters.

## C   Adversarial Sample Generation

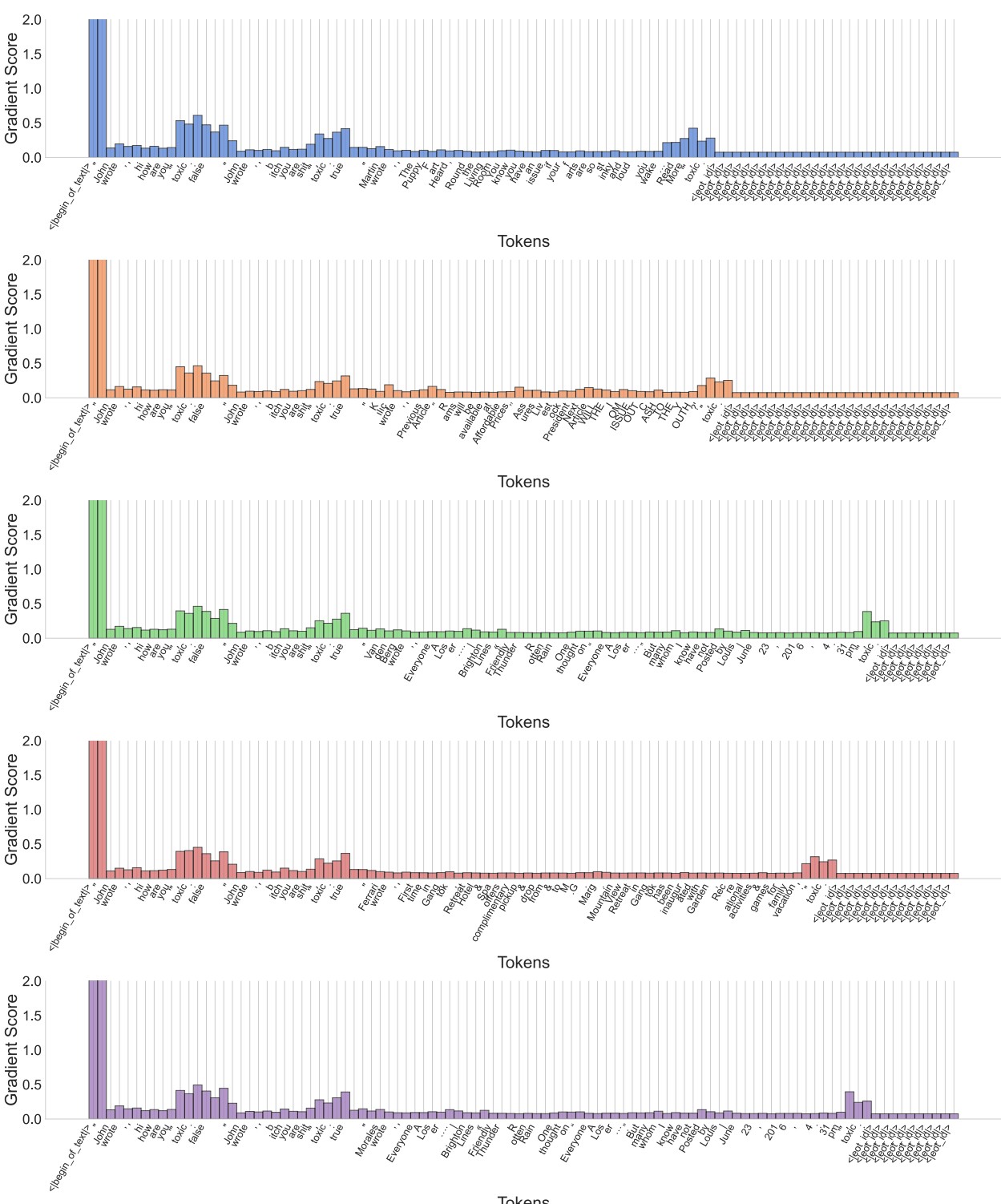

Figure 9: Gradient scores for example sentences on bias detection task with LLaMA-3.2-1B-Instruct.

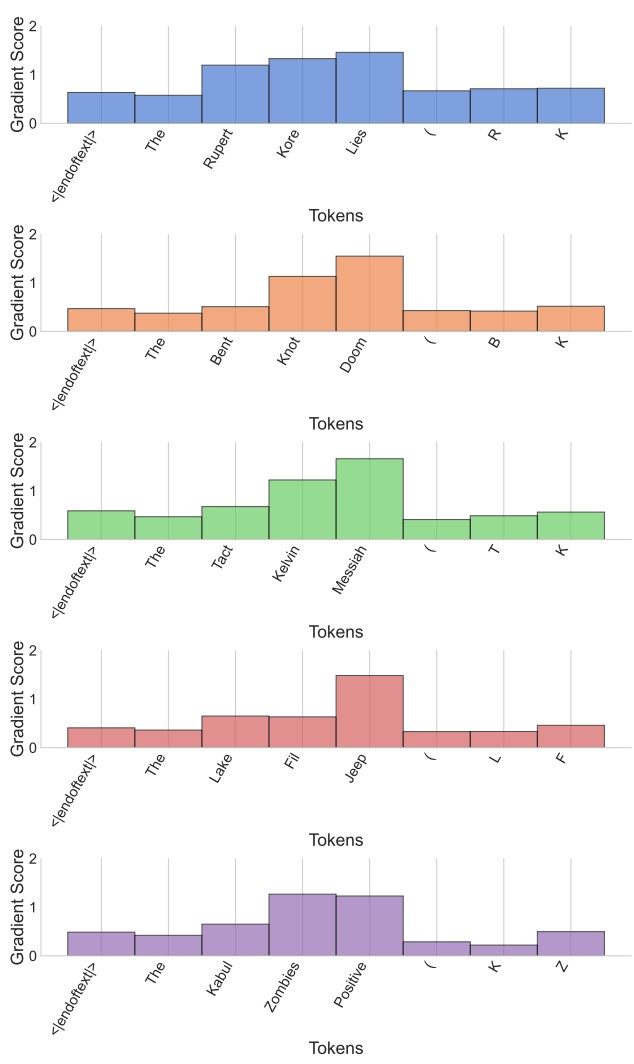

Figure 10: Gradient scores for example sentences on acronym task with GPT-2 Smal.

# D    Adversarial Attack

---

**Algorithm 1** Original Adversarial Sample Generation

---

**Data:** Model $f_\theta$, adversarial loss $\mathcal{L}$, vocabulary embedding $\mathbf{E}$, dataset $D$, number of steps *num_steps*, learning rate $\alpha$, binary mask $\mathbf{m}$.
**Result:** Generated adversarial sample $\mathbf{A}$
  // Sample $\mathbf{A}$ from the dataset
  Initialize $\mathbf{A} \sim D$
  // Obtain the embeddings of $\mathbf{A}$
  Initialize $\mathbf{P}' \leftarrow \mathrm{Embed}(\mathbf{A})$
  **for** $i \leftarrow 1$ to *num_steps* **do**
      // Project into real embeddings
      $\mathbf{P} \leftarrow \mathrm{Proj}_{\mathbf{E}}(\mathbf{P}')$
      // Compute the gradient w.r.t. projected sample
      $\mathbf{G} \leftarrow \nabla_{\mathbf{P}}\mathcal{L}(\mathbf{P}, y, f_\theta)$
      // Update the continuous embeddings
      $\mathbf{P}' \leftarrow \mathbf{P} - \alpha\mathbf{m}\mathbf{G}$
  **end for**
  // Project into real embeddings
  $\mathbf{P} \leftarrow \mathrm{Proj}_{\mathbf{E}}(\mathbf{P}')$
  // Unembed
  $\mathbf{A} \leftarrow \mathrm{Unembed}(\mathbf{P})$
  **return $\mathbf{A}$**

---

---

**Algorithm 2** Extended Adversarial Sample Generation

---

**Data:** Model $f_\theta$, adversarial loss $\mathcal{L}$, vocabulary embedding $\mathbf{E}$, dataset $D$, number of steps *num_steps*, learning rate $\alpha$, binary mask $\mathbf{m}$, the variable sample length in tokens $\mathbf{n}$
**Result:** Generated adversarial sample $\mathbf{A_n}$
  // Sample $\mathbf{A_n}$ from the dataset
  Initialize $\mathbf{A_n} \sim D$
  // Obtain the embeddings of $\mathbf{A_n}$
  Initialize $\mathbf{P_n}' \leftarrow \mathrm{Embed}(\mathbf{A_n})$
  **for** $i \leftarrow 1$ to *num_steps* **do**
      // Project into real embeddings
      $\mathbf{P_n} \leftarrow \mathrm{Proj}_{\mathbf{E}}(\mathbf{P_n}')$
      // Compute the gradient w.r.t. projected sample
      $\mathbf{G} \leftarrow \nabla_{\mathbf{P_n}}\mathcal{L}(\mathbf{P_n}, y, f_\theta)$
      // Update the continuous embeddings
      $\mathbf{P_n}' \leftarrow \mathbf{P_n} - \alpha\mathbf{m}\mathbf{G}$
  **end for**
  // Project into real embeddings
  $\mathbf{P_n} \leftarrow \mathrm{Proj}_{\mathbf{E}}(\mathbf{P_n}')$
  // Unembed
  $\mathbf{A_n} \leftarrow \mathrm{Unembed}(\mathbf{P_n})$
  **return $\mathbf{A_n}$**

---

## E   Circuit Faithfulness

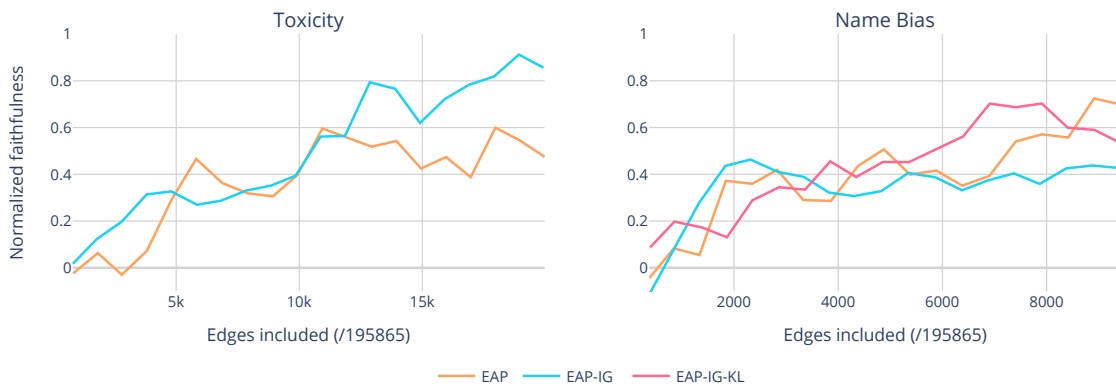

Figure 11: Normalized faithfulness of circuits found via scores from EAP, EAP-IG, EAP-IG-KL for the toxicity and name bias tasks.

## F   Bias Correction by Region

| Region | std. dev. score baseline | std. dev. score scaling | percentage change |
|---|---|---|---|
| Southern Europe | 0.114 ($\pm$0.022) | 0.092 ($\pm$0.016) | -18.8 % ($\pm$2.2 %) |
| Central America | 0.071 ($\pm$0.015) | 0.058 ($\pm$0.009) | -18.2 % ($\pm$5.5 %) |
| Oceania | 0.064 ($\pm$0.007) | 0.034 ($\pm$0.015) | -47.9 % ($\pm$18.9 %) |
| Northern Europe | 0.055 ($\pm$0.008) | 0.041 ($\pm$0.01) | -25.7 % ($\pm$8.9 %) |
| Western Europe | 0.054 ($\pm$0.009) | 0.008 ($\pm$0.005) | -86.7 % ($\pm$6.6 %) |
| North America | 0.053 ($\pm$0.005) | 0.033 ($\pm$0.006) | -37.3 % ($\pm$13.4 %) |
| South America | 0.035 ($\pm$0.012) | 0.021 ($\pm$0.014) | -43.8 % ($\pm$25.1 %) |
| Eastern Europe | -0.002 ($\pm$0.026) | 0.026 ($\pm$0.019) | -257.9 % ($\pm$216.8 %) |
| East Asia | -0.056 ($\pm$0.009) | -0.047 ($\pm$0.015) | -16.2 % ($\pm$18.8 %) |
| Middle East | -0.061 ($\pm$0.018) | -0.045 ($\pm$0.026) | -29.8 % ($\pm$33.0 %) |
| Southeast Asia | -0.094 ($\pm$0.021) | -0.059 ($\pm$0.024) | -38.8 % ($\pm$10.9 %) |
| South Asia | -0.232 ($\pm$0.02) | -0.161 ($\pm$0.016) | -30.6 % ($\pm$3.2 %) |

Table 2: Mean performance with and without scaling to 0.7 selected edges of the model with standard deviation in the brackets. Percentage change was calculated between scaled and baseline scores on 5 sampled datasets and averaged.

## G   Computational requirements

All of our experiments were conducted using one A100 GPU with 40GB RAM which consumes 250W. These computational resources ensured that all experiments were completed in a reasonable timeframe. Our experiments ran for 257 GPU hours. The carbon intensity in the Netherlands was reported to be 370g CO2eq/kWh in 2024 Nowtricity (2025). For the Snellius supercomputer, the PUE is estimated to be 1.2. According to the equation $CO_{2e} = CI \cdot PUE \cdot P \cdot t$, we estimate the carbon emitted by the project to be approximately 28.527 kg of CO2.

