# OpenReview forum: "Debiasing Through Circuits: A Reproducibility Study in Mechanistic Interpretability"
_TMLR — Rejected by TMLR_

### Review · Reviewer_aQEa · 2025-03-23

**Summary Of Contributions:**

This paper focuses on tools for identifying the components of an LLM responsible for a particular behavior and editing them. They use existing tools (activation patching, gradient-based adversarial attacks, and logit attribution) plus edge attribution to study toxic name-relaed biases in Llama-3.2-1B-Instruct. Overall, the paper's reads as 3/4 case-study and 1/4 and introduction of DICE. They offer a key demo of how the same method can fail to identify the same circuitry under changes to the dataset which is useful.

Overall, I think that this paper is good and that it would not be a mistake to accept to TMLR. However, I have thoughts/concerns below about the practical usefulness of this work. I feel relatively on the fence overall.

**Audience:**

Yes

**Broader Impact Concerns:**

None.

**Claims And Evidence:**

Yes

**Requested Changes:**

C1: A cite of Jain et al. (2024) is a citet when it should be a citep.

C2: Fig 2 should have larger font

**Strengths And Weaknesses:**

S1: I think that the writing and clarity are publication-quality. Figure captions could be made a little more descriptive though. Ideally, they would be self-explanatory without requiring reference to the text. I also wish that claims 1-3 were summed up more concisely somewhere in bold. The bold heads in section 5.2 would also potentially be better as complete sentences than just labels.

S2: Making tools that sorta work at the 1B scale for a non toy task seems to be pretty impressive compared to a lot of mechinterp work. One can reasonably be pessimistic about how this is not yet a scale and task that would make DICE useful for SOTA LLMs, but this paper still deserves some flowers over this.

S3: I think that the experimental approach and results about the difficulty of using interp to find adversaries is good.

W1: I think that the main challenge with this paper is similar to many other mechanistic interpretability papers. The paper isn't really grounded in solid evidence that the interpretations made or the methods introduced are valuable for practical tasks. For example, DICE is a cool method, and Figure 7 is nice. But ideally, we would want to see how good DICE is compared to a baseline like fine-tuning.

N1: One thing that I don't think is a weakness of this paper that it builds on Garcia-Carrasco et al. (2024). I think that this paper both scales things up and adds significant substance such that I don't think that a lack of novelty is a weakness.

---

> ### Author Response · Authors · 2025-04-12
> **Addressing review**
>
> Thank you for reviewing our paper and for the thoughtful feedback. We have addressed the requested changes (C1,C2).
> > I think that the main challenge with this paper is similar to many other mechanistic interpretability papers. The paper isn't really grounded in solid evidence that the interpretations made or the methods introduced are valuable for practical tasks. For example, DICE is a cool method, and Figure 7 is nice. But ideally, we would want to see how good DICE is compared to a baseline like fine-tuning.
>
> We appreciate this concern and would like to clarify the scope and intent of our submission. We intend this paper to be part of the ML Reproducibility Challenge 2025 under TMLR, with the primary goal of reproducing the findings of Garcia-Carrasco et al. (2024) (we added justification for why the prior work needs to be reproduced based on the feedback from Reviewer WZ7W). Accordingly, most of our analysis and commentary is centered on replicating and critically assessing that work.
> The introduction of the DICE method is intended as a natural extension, building on the circuits comparisons we made in the paper. While DICE shows promise, we agree that a thorough evaluation across different models and tasks would be necessary to fully establish its generalizability and practical utility. We believe DICE is a valuable direction that deserves a standalone paper with deeper empirical grounding, and we appreciate the suggestion to benchmark it more directly against baselines like fine-tuning.

---

> > ### Comment · Reviewer_aQEa · 2025-04-12
> > **thanks**
> >
> > Thanks, for the reply. Let me know if I am out of line here, and the area chair can override me. I think this paper does a really good job of critically analyzing and building on. Garcia-Carrasco. But I don't think this does much to change my main concerns. I don't see this much as a substantive contribution in and of itself. The kind of thing that would convince me to write this paper more highly would be to scale things up significantly and or to beat other model intervention, baseline such as fine-tuning or representation engineering methods.

---

### Review · Reviewer_WZ7W · 2025-03-28

**Summary Of Contributions:**

This paper considers the problem of using mechanistic interpretability to explain why models are susceptible to adversarial examples. The paper primarily reproduces the results from a prior work that addresses the same problem. Then the paper makes some small tweaks to the technique proposed in the prior work and applies it to analyze a larger model (Llama-3.2-1B-Instruct) on a different task. Finally, the paper also presents a simple method for editing models to remove undesirable behavior.

**Audience:**

Yes

**Claims And Evidence:**

Yes

**Requested Changes:**

The main change needed is a proper justification for why the prior work needs to be reproduced. Since this new work does not have any new technical contributions (except the model editing approach), it is really important to make a strong case for reproducing the previous paper. Otherwise, the purpose of this paper is not clear.

I also list some clarification questions that can be addressed by including appropriate clarifications in the paper.

### Clarification Questions:
1. Sec 3.2: "The binary target of each datapoint is derived from the toxicity score" --> How?

2. Sec 3.3: Is resampling ablation performed separately for each node? If so, what does "circuit" correspond to in Equation (3)? It would help to explain precisely how circuit faithfulness is computed? Also, the explanation of the EAP-IG method is not very clear. I could not understand what the method does from the text in this paper.

3. Sec 3.4: It is not at all clear to me how the adversarial example generation algorithm in this paper is different from prior work. The section mentions "multitoken adversarial generation" but I am not sure what this means. Couldn't the previous algorithm also modify multiple tokens in the input?

4. Sec 3.6: What is "z-normalized logit difference"?

5. Sec 4, "Experiment 4": It is not clear what is the new method (from Sec 3.3) that is used to find the vulnerable attention heads. Is it just EAP? How is this different from circuit discovery?

6. Sec 5.2, "Circuit Identification using EAP": Is there some explanation for why EAP is better than activation patching? How is the score for a head calculated using EAP?

7. Sec 5.2, "Gradient Flow": It is not clear from the section whether the algorithm was successful in finding adversarial examples or not. Please report on how many successful adversarial examples were found?

8. Sec 5.3, "Finding Vulnerable Components": "The plotted circuit is significantly smaller ...." --> Which circuit is this referring to? Isn't the aim to find vulnerable heads? Why is this section then talking about circuit discovery?

9. Sec 3.2 and Sec 5.2: Which data split is used for the experiments? Why is data split 5 ways in the first place? Why not use all data for the experiments? It would be helpful to more carefully describe the data usage for each of the experiments.

**Strengths And Weaknesses:**

### Strengths
1. The paper brings together a number of methods such as edge attribution patching, resampling ablation, generating adversarial examples, and logit lens to mechanistically interpret model weaknesses. However, I note that actually it is the prior work, that the paper reproduces, which brought all these methods together. This paper replaces some of the methods used in the prior work such as attribution patching with newer methods such as edge attribution patching.

2. The use of mechanistic interpretability methods to detect biases in models and to correct them is interesting.

### Weaknesses
1. The paper is primarily a reproducibility study with no new technical contributions except the method to edit the models. I am not sure why such a reproducibility study is even necessary and I do think it helps us learn anything significantly new.

2. The paper is not written in a self-contained manner and refers too often to the past work being reproduced. It almost feels like that the two papers need to be read together. I would do this if I was given a good justification for why the past work was important and why it needs to be reproduced. However, the paper does not provide any such justification.

---

> ### Author Response · Authors · 2025-04-12
> **Addressing review**
>
> Thank you for the review and in-depth analysis of our paper, thanks to your comments we were able to improve the quality of our submission.
>
> >The main change needed is a proper justification for why the prior work needs to be reproduced.
>
> We have justified the importance of reproducing the original paper at the end of Section 1. Please note that this submission is intended for the ML Reproducibility Challenge 2025 under TMLR, with the primary goal of reproducing the findings of Garcia-Carrasco et al. (2024).
>
> We address each clarification question individually:
>
> 1. The additional explanation about the derivation of the binary score has been added in section 3.2.
> 2. We updated section 3.3 and added Appendix A - Mechanistic Interpretability to address these questions.
> 3. As we highlight in the updated section 3.4, the original algorithm was not able to generate multi-token adversarial samples out of the box. The main part of the algorithm remains the same, but the real (token space) embeddings chosen by Eq. 4 are different.
> 4. The technique has been explained in section 3.6.
> 5. We clarified the new method in Experiment 4. The difference between activation patching and circuit discovery is explained in Section 3.3. EAP is a method for circuit discovery.
> 6. We added some necessary explanations in Section 3.3, which were later used to explain how head scores are calculated in Section 5.2: Circuit Identification using EAP.
> 7. We updated Section 5.2: Gradient Flow with numerical values and discussion.
> 8. We have revised the phrasing in Section 5.3: Finding Vulnerable Components to improve clarity and avoid potential ambiguity. In this section, vulnerable heads are identified by summing the normalized edge scores for heads and layers within the discovered circuit.
> 9. We believe that splitting the data enables a more robust performance analysis while also providing greater flexibility in selecting dataset sizes for more computationally expensive experiments. We have added an explanation in Section 3.2. For each experiment in Section 5.2, we now explicitly report the number of splits used.

---

> > ### Comment · Reviewer_WZ7W · 2025-04-23
> >
> > Thank you for the response and the clarifications! However, I echo the concern of the other reviewers. While I see the value of the proposed DICE method, I am remain unconvinced of the need to reproduce the results of Garcia-Carrasco et al. (2024).

---

### Review · Reviewer_gt1o · 2025-04-09

**Summary Of Contributions:**

The paper extends a mechanistic interpretability pipeline to identify and mitigate bias in LLMs by introducing a novel method called Differential Circuit Editing (DICE). Experiments show that the authors' approach reduces bias and improves accuracy in toxicity detection.

**Audience:**

Yes

**Broader Impact Concerns:**

This work takes a meaningful step toward making language models fairer and more trustworthy by showing that it's possible to reduce bias without hurting performance. This is a positive broader impact. I could not identifying any potential concerns related to a broader impact of the research.

**Claims And Evidence:**

No

**Requested Changes:**

Please see comments above.

**Strengths And Weaknesses:**

The paper presents a moderately novel extension to existing mechanistic interpretability methods. Their contribution makes prior approaches more scalable and allows a targeted approach to bias mitigation in LLMs.
The notation in the paper is generally clear and consistent, particularly in the formalization of circuit interventions and metrics like logit difference and bias scores.
The exclusive focus on a single model architecture (LLaMA-3.2-1B-Instruct) makes me somewhat skeptical about the generalizability of the findings, which is one of the major claims. Particularly the effectiveness of DICE and faithful circuit identification. It would strengthen the work to validate these methods across diverse model families.
Could you please clarify how circuit faithfulness thresholds were selected and whether the edge attribution methods yield consistent circuits across seeds or prompt variations, as these are key for assessing reproducibility?
If I understand it correctly, gradient-based adversarial attacks fail in more complex tasks, yet authors still rely on them. Consider integrating or comparing with alternative attack methods.

---

> ### Author Response · Authors · 2025-04-12
> **Addressing review**
>
> Thank you for reviewing our paper and for the thoughtful feedback.
>
> > It would strengthen the work to validate these methods across diverse model families.
>
> We agree that it would strengthen the work to validate the methods across diverse model families. However, we would like to clarify the scope and intent of our submission: this work is intended for the ML Reproducibility Challenge 2025 under TMLR, with the primary objective of reproducing the findings of Garcia-Carrasco et al. (2024). Existing MI research typically uses small models (specifically GPT2-small). There are a few works that scale their experiments to larger models. However, to our best knowledge, our paper is one of the few that tests its findings on a non-trivial task with large-scale model. We chose Llama3.2-1B-Instruct explicitly because it comes from an open-source family of models, which represents the largest model we could run.
>
> This work is not meant to be an extensive study of DICE and faithful circuit identification. The introduction of the DICE method is intended as a natural extension, building on the circuits comparisons we made in the paper. While DICE shows promise, we agree that a thorough evaluation across different models and tasks would be necessary to fully establish its generalizability and practical utility. We believe DICE is a valuable direction that deserves a standalone paper with deeper empirical grounding. Faithful circuit identification is well studied by the mechanistic interpretability community (we cite Hanna et al. (2024), Miller et al. (2024)), so we chose it as a well-grounded method for comparison with activation patching used by Garcia-Carrasco et al. (2024).
>
>
> >Could you please clarify how circuit faithfulness thresholds were selected and whether the edge attribution methods yield consistent circuits across seeds or prompt variations, as these are key for assessing reproducibility?
>
>
> We updated Section 3.3 to explain the threshold selection. EAP methods are sensitive to prompt variations - differences would result in different edge scores which would not yield exactly the same circuits. Consistency is a known issue in circuit discovery, but using samples from the same data distribution should result in the same core components of the identified circuit. We provide the necessary specifications to replicate our circuits.
>
>
> >If I understand it correctly, gradient-based adversarial attacks fail in more complex tasks, yet authors still rely on them. Consider integrating or comparing with alternative attack methods.
>
>
> We updated the Section 5.2: Gradient Flow with an explanation. Essentially, the experiment was designed to show the ineffectiveness of the method when scaled to a harder task.

---

> > ### Comment · Reviewer_gt1o · 2025-04-24
> > **answer**
> >
> > Dear authors, thanks for your response and clarification. If the goal was to reproduce the original work, then I think the paper succeeds. But in my view, a strong reproduction should also bring some added value. The diirection the authors propose might be promising, but at this point, I still feel to echo others reviewer in saying the experimental scope feels too limited to fully support the claims or show broader impact.

---

### Author Response · Authors · 2025-04-26
**Rebuttal**

We would like to sincerely thank the reviewers for their constructive feedback, which has helped us to improve the clarity and positioning of our work. In this rebuttal, we would like to clearly restate the motivations behind reproducing the specific paper and articulate the contributions our study brings beyond mere reproduction.
By writing this paper, we aimed to answer several questions that we believe are not sufficiently addressed in the current literature:
* Can methods that detect model internals responsible for specific behaviors be effectively applied to critical domains such as fairness and bias in large-scale models?
* How does a head-independent method, such as projecting logit differences, compare to more comprehensive model inspection techniques like full circuit analysis approaches, for example edge attribution patching (EAP)?
* Can the idea of selectively disabling specific edges within a model to mitigate undesired behaviors be generalized and practically applied to large, publicly available models?

The original paper we chose to reproduce, Garcia-Carrasco et al. (2024), which focuses on Model Interpretability (MI) and particularly on adversarial example generation, was selected because it aligns closely with the gaps we intended to address:
* The original work introduces a method for obtaining adversarial samples that can be highly useful in fairness and bias studies, an increasingly important area of research.
* It integrates several widely-used MI methods to identify components responsible for unwanted model behavior, making it a practical tool for broader applications.
* However, the original evaluation was limited to toy datasets, leaving open the question of its scalability and real-world utility.
* The original method does not incorporate full circuit analysis approaches like EAP, which opens an interesting opportunity for comparative evaluation.

Beyond reproducing the original results, our paper brings the following contributions:
* We investigate the comparison between complete circuit discovery methods and approaches focused on selective components (e.g., attention heads) for identifying responsible mechanisms inside models.
* We create a new dataset and conduct a systematic study focused on fairness and bias tasks, which are currently critical topics, especially as large language models (LLMs) are trained on ever-larger and potentially biased datasets.
* We apply the method at scale to large, publicly available model, testing its effectiveness and robustness in more realistic settings compared to the toy example originally used.
* We validate the adaptability and versatility of disabling model edges to isolate and mitigate specific behaviors, demonstrating its application on large-scale bias mitigation tasks.

In summary, while reproducing the original method on toy examples was not the primary focus, our work builds upon it by integrating both the original techniques and broader circuit analysis methods such as EAP, significantly extending the evaluation to large-scale models and important domains like fairness and bias.

---

### Decision · Action_Editor_AB6L · 2025-05-20

**Recommendation:** Reject

**Comment:**

Thank you for submitting your paper to TMLR. I appreciate the motivation to reproduce Garcia-Carrasco et al., however, the paper falls short of a reproducibility study that provides sufficient general insights. As reviewers noted, experiments beyond a single model may help in communicating general insights.

My other comment is that the paper is trying to achieve two goals: 1) changes to the original method to develop a better, scalable method; 2) reproducibility of the original study. If reproducibility is the goal, I would suggest the authors to do a more extensive experimentation setup that allows for generalizable insights. Or they may choose to focus on the novel improvements and aim to show benefits in practical settings.

I would encourage you to consider the reviewers' suggestions for a future version of the paper.

**Audience:**

This is a reproducibility study, but falls short of providing general insights since the experiments are on a single model and specific configuration. A study where multiple models and settings may be studied, may provide more generalizable insights to the TMLR audience.

**Claims And Evidence:**

The paper reproduces the results of a prior mechanistic interpretability study.
* Reproducibility: Yes, with some changes, the technique reproduces to a larger model.
* Scaling of the method: Results are presented on a LLama 1B model, bigger than the prior study.

However, the claims could be better supported if the experiments utilized more than one language model.